# Exploring Musculoskeletal Complaints in a Needle Manufacturing Industry: A Cross-Sectional Study

**DOI:** 10.3390/ijerph21080996

**Published:** 2024-07-29

**Authors:** Paulo C. Anacleto Filho, Ana Cristina Braga, Paula Carneiro

**Affiliations:** ALGORITMI Research Center/LASI, University of Minho, 4800-058 Guimarães, Portugal; acb@dps.uminho.pt (A.C.B.); pcarneiro@dps.uminho.pt (P.C.)

**Keywords:** physical ergonomics, musculoskeletal symptoms, cross-sectional study, manufacturing industry, working population

## Abstract

Musculoskeletal disorders (MSD) encompass a variety of conditions affecting muscles, joints, and nerves. In Portugal, MSDs are the most prevalent occupational health problem in companies. Based on the relevance of work-related MSD (WMSD), this study aims to assess the prevalence of MSD complaints in a needle manufacturing industry in Northern Portugal, following a cross-sectional approach. Thus, 526 workers from five departments (i.e., operator, tuning, maintenance, administration, and logistics) answered a sociodemographic questionnaire and the Nordic Musculoskeletal Questionnaire (NMQ). Within the last 12 months, females exhibited a higher frequency of complaints than males across all body parts except for ankles/feet. The body parts eliciting the most percentage of complaints for both genders include the lower back (54.2%), neck (42.2%), shoulders (39.0%), ankles/feet (38.2%), and wrists/hands (35.7%). No significant association was found between Body Mass Index (BMI) and body part complaints. Tuners reported the highest complaint rate, with occupations as substantial predictors of complaints in certain body parts. Likewise, complaints tend to increase with age. The findings advocate for ergonomic interventions that are gender-, age-, and job-sensitive.

## 1. Introduction

Musculoskeletal disorders (MSD) are a broad range of health problems that affect the body’s connective, muscular, and nervous tissues. These conditions result from various factors, including inflammatory diseases, age-related changes, and levels of physical activity [1]. It is possible to categorise MSD into two distinct spheres: work-related musculoskeletal disorders (WMSD) and those arising from non-work-related factors. This classification underscores the diverse risk factors for MSD, encompassing lifestyle, individual, and workplace factors [2,3]. Among workplace factors, one can cite repetitive movements, forceful exertions, and awkward postures [3]. Moreover, the impact of physical agents, such as vibrations [4] and exposure to cold [5], holds considerable significance in WMSD research. Other risk factors that can contribute to the development of WMSD are psychological [6], psychosocial [7] and organisational nature [8]. Additionally, the cumulative nature of MSD development is crucial, as it typically emerges from exposure to several risk factors over time. Within this study, musculoskeletal complaints serve as potential indicators of underlying MSD, encompassing self-reported experiences of pain, discomfort, or limitations in movement. Monitoring the prevalence of complaints can function as an initial step in identifying potential MSD within working populations.

Extensive research has been conducted on musculoskeletal symptoms, within which various critical lines of investigation stand out. Notably, researchers have examined the body segments most affected by different tasks and working conditions. For instance, studies have delved into occupational and non-occupational factors associated with troubles in the back [9,10], cervical region [11], feet [12], hip [13], knee [14], shoulder [2], and wrist [15]. Additionally, research has explored how sociodemographic factors can influence the occurrence of WMSD [16] and the impact of psychosocial, psychological [17], and organisational factors on workers’ health [8].

Studies have also focused on specific industries. For instance, Gomes et al. [18] and Souza et al. [19] conducted comprehensive assessments of WMSD risk factors among 420 workers from trade, education, industry, and health sectors. Likewise, Wang et al. [20] scrutinised the prevalence and associated risk factors of WMSD, encompassing 1415 workers within an industrial cluster in China. Focusing specifically on manufacturing industries, Lu et al. [21] evaluated the risk of WMSD among 393 operators in electronic manufacturing, while Chu et al. [22] examined the interplay of personal and work-related factors contributing to shoulder MSDs in 931 electronic industry workers. Yang et al. [23] furthered the understanding of WMSD relations with work-related variables by investigating a cohort of 7307 workers in the electronics sector. Maimaiti et al. [11] explored the associations between cervical MSD prevalence and individual, physical, and psychosocial factors in a study with 700 assembly workers. Lastly, Hembecker et al. [24] explored the metal products manufacturing industry, elucidating the links between MSD symptoms and sociodemographic and occupational characteristics based on an analysis of 226 workers.

Furthermore, there has been an emphasis on the development of hardware and software tools to evaluate ergonomics risks in relation to MSDs [25], along with inquiries into the influence of ageing on MSDs [26]. This substantial interest in MSDs stems from their far-reaching impacts on individuals, organisations, and society. At the individual level, MSD has a significant effect on workers’ well-being [1,17], whereas at the organisational level, they may impact business performance [27] through absenteeism, productivity loss, increased health care costs, disability, and worker’s compensation costs [1,28,29]. Furthermore, at the societal level, MSDs are one of the primary sources of public health expenditure in many countries (e.g., EUA [30] and EU [1,29]). In 2019, MSD accounted for 15% of the public expenses in the EU-28, falling behind expenses with cancer only [29]. Therefore, MSDs are the leading work-related health problem in the EU [31]. The associated financial impact is also substantial, as estimated by Bevan [28], who suggests that productivity losses linked to MSD among the working-age population in the EU could reach up to 2% of the gross domestic product (GDP) (while all references cited within this paragraph address Musculoskeletal Disorders (MSDs), the specific definitions employed may vary. For a comprehensive understanding of these nuances, readers are directed to consult the individual references).

In Portugal, according to the 2015 National Survey on Working Conditions Survey, MSDs are the most prevalent occupational health problem in companies [32]. Based on the relevance of MSDs, this study aims to explore the relationship between MSD complaints and sociodemographic and biomechanical risk factors in a specific industry in Northern Portugal, following a cross-sectional approach [33]. This industry manufactures components for the textile industry, specifically sewing needles. To the authors’ knowledge, this is the first study in the peer-reviewed scientific literature to assess this industry regarding MSD complaints. Within this case study, 526 workers from five departments (i.e., operating, tuning, maintenance, administration, and logistics) answered a sociodemographic questionnaire and the Nordic Musculoskeletal Questionnaire (NMQ).

## 2. Materials and Methods

### 2.1. Study Design

This cross-sectional study utilised Strengthening the Reporting of Observational Studies in Epidemiology (STROBES) recommendations to strengthen its credibility and replicability [34]. This research was conducted within a manufacturing company located in the Northern region of Portugal. This company primarily focuses on producing components for the textile industry (i.e., sewing needles). In adherence to principles of confidentiality, the company’s name is deliberately withheld from disclosure in this paper. The company had 644 employees when the questionnaires were applied (i.e., target population).

Each employee adheres to an 8 h daily workload from Monday to Friday. The work schedules are categorised into rotating and fixed shifts with a 30 min break for lunch or dinner. In the case of rotating shifts, employees alternate their schedules weekly. For example, one week involves working from 06:00 to 14:30, followed by the next week’s schedule from 14:30 to 23:00. Meanwhile, fixed hours encompass both day and night shifts.

### 2.2. Data Collection

The standardised NMQ in its Portuguese version [35] was applied to assess the prevalence of musculoskeletal symptoms among participating workers. Each participant who agreed to participate in this study did so voluntarily after being informed about the research purpose, methods, and the anonymity of the obtained data. Participants were also informed that they could leave or oppose the data collection process at any time. Inclusion criteria required participants to be at least 18 years old and employed for at least 12 months.

Only responses referring to the question related to experiencing trouble (ache, pain, discomfort, numbness) in the previous 12 months were analysed, as in [36] and [23]. Furthermore, a sociodemographic questionnaire was applied to collect data on age, gender, years of service in the same job, work shift, etc. Both questionnaires were distributed on paper to the target population (644 employees). Table 1 summarises the information collected through the questionnaires. The data collection took place over three months, from January to March 2018. The collected responses amounted to 537 eligible participants, but only 526 questionnaires were suitable for analysis, as 11 were incomplete, resulting in a response rate of 81.7%.

### 2.3. Data Analysis

All collected data was stored in Excel 2019 (Microsoft^®^). The data analysis process was performed using SPSS Statistics software version 29 (SPSS Inc., Chicago, IL, USA) and Excel 2016 (Microsoft^®^).

According to the central limit theorem, the assessment of data normality was omitted due to the substantial sample size (n = 526) as recommended by [37,38]. To compare and describe data, averages (AVG), standard deviations (SD), and frequencies, were employed. Differences in age and BMI were assessed using the independent *t*-test (with a 95% confidence level). In the t-test, *p*-values below 0.05 (*p*-value < 0.05) denote statistical significance in the compared variables, while *p*-values exceeding 0.05 (*p*-value > 0.05) indicate equality between compared variables [38].

The Chi-square test (χ^2^) was employed to examine the association between two categorical variables. In this test, *p*-values under 0.05 (*p*-value < 0.05) mean a statistically significant association between variables, whereas *p*-values above 0.05 (*p*-value > 0.05) mean a lack of association between variables [37,38]. Upon identification of a statistically significant correlation, the magnitude and direction of this association were quantified using Cramer’s V. A Cramer’s V effect size smaller than 0.2 is considered weak, whereas from 0.2 to 0.6 moderate, and above 0.6 is considered strong [39].

When conditions for the Chi-square test were not satisfied (see [38]), Fisher’s exact test was utilised for 2 × 2 contingency tables to ascertain the association between variables. For contingency tables exceeding the dimensions of 2 × 2, Pearson’s Chi-square test was applied. The odds ratio and the relative risk were computed to identify possible trends between categorical variables.

All associations involved all 526 respondents except for associations involving the Logistics occupation and BMI. Due to the small number of individuals in the Logistics homogeneous group of exposure (1.5% of the total sample), this group was excluded from occupation-related associations (Section 4.2.3), resulting in a total of 518 individuals.

Data regarding BMI were available for 444 participants (84.4% of the total sample) who responded to weight and height inquiries. Similarly, age data were obtainable for 437 participants (83.1% of the total sample).

## 3. Results

This section presents results concerning the sample’s sociodemographic characteristics (Section 3.1), followed by the distribution of complaints across gender (Section 3.2), age (Section 3.3), occupation (Section 3.4) and BMI sample characterisation (Section 3.5).

### 3.1. Sociodemographic Characteristics

The final sample comprised 526 workers, representing 81.7% of the company’s 644 workers. Concerning gender composition, the final sample comprised 30 females (5.7%) with an average age of 45.9 ± 10.3 years and 496 males (94.3%) with an average age of 40.0 ± 18.0 years. It was observed that females were older than their male counterparts [t (484) = −9.963; *p* < 0.001)]. The average weight was 67.7 ± 13.3 kg for females and 80.4 ± 12.1 kg for males. Additionally, the average stature for females was 1630.7 ± 68.3 mm, and for males, it was 1748.5 ± 74.2 mm. The average BMI was 25.3 ± 5.0 for females and 26.3 ± 3.62 for males. Regarding dominant handedness, 80.0% of females and 92.1% of males were right-handed. Table 2 shows the gender-based distribution of workers across different occupations.

Regarding working shifts, 90.0% of females and 29.6% of males worked fixed shifts, while the remainder worked rotating shifts. The average years of service in the same workstation was 16.5 ± 11.2 years. When considering the effect of gender, the average years of service for females was 20.0 ± 12.0 years and 16.3 ± 11.8 years for males. Figure 1 illustrates the distribution of individuals across different ranges of years of service.

To highlight the representativeness of the current sample for the entire needle manufacturing industry in Portugal, the following economic activities were considered: “Basic metallurgical industries” (CAE Rev. 3 activity 24) and “Manufacturing of metallic products, except machinery and equipment” (CAE Rev. 3 activity 25). Historical statistical data from 1995 to 2021 show that these two economic activities employed an annual average of 97,500 workers [40]. According to Israel [41], a sample size of 398 workers is necessary to achieve a 5% precision level for a population of 100,000 with a 95% confidence level. Thus, the current sample size can reliably represent the broader needle manufacturing industry in Portugal.

### 3.2. Gender-Related Complaints

Regarding body part complaints within the last 12 months, females exhibited a higher frequency of complaints compared to males across all body parts except for ankles/feet (Figure 2). The body parts eliciting the most percentage of complaints for both genders include the lower back (54.2% of the sample), neck (42.2%), shoulders (39.0%), ankles/feet (38.2%), and wrists/hands (35.7%). Figure 2 illustrates the distribution of complaints per body part and gender.

Among females, the body part with the highest number of complaints was the lower back (66.7%), followed by the neck, shoulders, and wrists/hands (53.3% each), knees (36.7%), elbows and hip/thighs (26.7% each), ankles/feet (23.3%), and upper back (13.3%). In contrast, in males, the most affected body parts were the lower back (53.4%), followed by the neck (41.5%), ankles/feet (39.1%), shoulders (38.1%), wrists/hands (34.7%), knees (34.1%), hip/thighs (22.0%), elbows (15.5%), and upper back (7.1%).

### 3.3. Age-Related Complaints

Females exhibited the highest complaint rate per individual across all age groups, except for the 39–48 age group. Additionally, the number of complaints per individual increased with age for males, from 1.2 in the 19–28 age group to 3.7 in the 49–58 age group, subsequently declining to 2.9 in the 59–68 age group. In contrast, females exhibited a steady number of complaints up to the 39–48 age group, then increasing from 4.3 in the 49–58 age group to 5.8 in the 59–68 age group. For females, the highest complaint rate per individual was in the 59–68 group age and for males was in the 49–58 group age. These results, depicted in Figure 3, indicate a difference in the complaint rates across different age groups between genders.

### 3.4. Occupation-Related Complaints

Table 3 presents employees’ tasks, categorised by the homogeneous exposure groups and associated biomechanical risk factors. When evaluating the number of complaints relative to specific body parts across various occupations, it was observed that Tuners presented the highest complaints rate with 3.3 complaints/worker. This was followed by Maintenance with 3.2 complaints/worker, Administrative with 3.0, Operators with 2.3, and Logistics with 2.1. Figure 4 shows the percentage of complaints per body part and occupation.

When assessing occupation-related complaints by gender, females demonstrated the highest percentage of complaints in the operator’s role, representing 43.3% of all reported issues. This was followed by Administrative roles, where 38.9% of workers reported complaints, and Logistics roles (22.2%). In contrast, among males, the highest frequency of complaints was reported by Tuners, with an average of 36.5% of all reported issues. This was followed by Maintenance roles, where 35.6% of male workers reported complaints, Administrative roles (31.5%), Operators (24.9%), and Logistics (24.1%). Figure 5 and Figure 6 depict the distribution of complaints across body parts and occupations for females and males, respectively.

Additionally, it was observed that female operators reported a higher frequency of complaints in all body parts except ankles/feet than their male counterparts. The most affected body part among female Operators was shoulder (80.0%), followed by lower back (70.0%), neck and wrists/hands (60.0% each), elbows (40.0%), knees (30.0%), ankles/feet and hips/thighs (20.0% each) and upper back (10.0%). For male Operators, the highest frequency of complaints was lower back (43.9%) followed by shoulder (32.4%), neck (31.8%), ankles/feet (29.5%), wrists/hands (28.3%), knees (22.0%), hips/thighs (18.0%), elbow (11.6%) and upper back (6.4%).

Based on the complaints of females and males in administrative roles, it was observed that females reported a higher frequency of complaints in all body parts except the elbows than their male counterparts. The highest frequency of complaints among females in Administrative roles was lower back (61.1%), neck (55.6%), wrists/hands (50.0%), shoulder and knees (44.5% each), hips/thighs (33.3%), ankles/feet (27.8%), elbows and upper back (16.7%). Among males in Administrative roles, the highest frequency of complaints was lower back (54.2%), neck (45.8%), shoulder (43.7%), knees (41.7%), wrists/hands (31.2%), hips/thighs (25.0%), ankles/feet and elbows (18.7% each) and upper back (4.2%). In terms of proportion, the body parts eliciting the most complaints for both genders in the Administrative role included the lower back, neck, shoulders, knees, and wrists/hands. Notably, the lower back and neck were the most frequently reported complaints for both genders.

Upon comparing the complaints from females and males in Logistics roles, it was observed that the lower back was the primary area of discomfort for both genders. This was followed by elbows and wrists/hands (50.0% each) for females and knees, ankles/feet and shoulders (33.3% each), neck, wrists/hands and hips/thighs (16.7% each) for males. Concerning complaints in wrists/hands and lower back, females registered a higher frequency of complaints than males. However, complaints related to the neck, shoulders, hips/thighs, knees, and ankles/feet were exclusive to males working in Logistics.

### 3.5. BMI

Considering the importance of BMI as a health indicator, available weight and height measures (n = 444) were employed to calculate it. Therefore, based on the BMI standard weight status categories [42], the BMI of males and females was classified as shown in Table 4. Accordingly, 4.2% of females and 0.2% of males were ‘underweight’, 50.0% of females and 38.3% of males were classified as ‘normal weight’, 29.2% of females and 49.5% of males were classified as ‘Pre-obesity’, and 16.7% of females and 11.9% of males pertained to ‘obesity’ categories (BMI ≥ 30.0). Statistically, no significant difference was observed between the BMI of males and females [t (447) = −1.274; *p* > 0.05)].

When stratified by age groups (Table 5), it was observed an incidence of overweight/pre-obesity (n = 62, 52%) and obesity (n = 19, 15.8%) in the age group 39–48. Interestingly, in all age groups older than 29 years old, the number of people being overweight or obese was higher than those under or normal weight.

## 4. Discussion

### 4.1. National Representativeness

This study’s sample of 526 workers closely mirrored the company’s gender distribution (females, 5.7% in this study vs. 6.0% in the company; males, 94.3% in this study vs. 94.0% in the company) and was well-representative of the company’s population (n = 526 in this study v. N = 644 in the company). However, male (94.3%) and female (5.7%) populations were not similar to the national labour force participation of 49.7% females and 50.3% males [43]. A series of factors, such as the organisational culture in the company, can explain this difference [44], the economic sector [45], and gender preferences [46].

According to the most recent data on the composition of the Portuguese workforce per age group, 6.2% are between 15–24 years old, 42.5% are between 25–44 years old, 27.4% are between 45–54 years old, 19.6% are between 55–64 years old, and 4.4% are 65 or more years old [47]. By way of comparison, the same age groups were considered below, with workers age distribution revealing that 5.7% of the company workforce are aged 15–24 years old, 56.1% 25–44 years old, 19.0% are between 45–54 years old, 17.1% are between 55–64 years old, and 2.1% are 65 or more years old (Figure 7). Consequently, the final sample represents the age distribution of the Portuguese workforce well.

### 4.2. Associations between Gender, Age, Job, and Body Part Complaints

#### 4.2.1. Gender vs Body Part Complaints

When evaluating body parts complaints about gender, it was observed that the only significant association (*p*-value ≤ 0.05) was wrists/hands, with a Cramer’s V effect size (0.090) indicating a weak association between variables (Table 6). According to the relative risk ratio, women were 1.54 more likely to have complaints in their wrists/hands than men. The odds ratio shows that the chances of complaints in this region were 2.15 times greater for females than males.

These results may be more influenced by individual factors like gender and lifestyle habits than workplace factors alone. Studies have shown that women are more susceptible to carpal tunnel syndrome (CTS) than men [15], a condition directly linked to hand and wrist issues. Hormonal fluctuations experienced by women, particularly during pregnancy, breastfeeding, and menopause, are known contributors to CTS development [15]. Notably, our sample’s higher average age for women (45.9 ± 10.3 years for females vs. 40.0 ± 18.0 years for males) suggests a potential link to these hormonal changes. Furthermore, repetitive forceful exertions, both recreational and domestic, alongside extreme wrist flexion, can contribute to dangerous stress on the wrist and hand complex [48].

Although the remainder of the variable associations were non-significant (*p*-value > 0.05) (Table 6), several works in the literature have already presented evidence for gender-based differences in the prevalence of pain or diseases across different body regions. For instance, Harris and Coggon [13] have shown that hip osteoarthritis rates were higher in females than males. Likewise, most studies indicate a higher prevalence of neck pain among women than men [11,49,50,51]. Also, Leboeuf-Yde et al. [52] and Shiri et al. [53] reported that females presented higher rates of lower back pain than males. Regarding foot complaints, Bhatia [54] highlighted that, on average, females are nine times more likely to have various foot complaints than males, one possible reason being inappropriate footwear. Concerning shoulders, a Finish cohort study carried out by Miranda et al. [55] showed that males developed shoulder disorders more often than females.

Nonetheless, even though the only significant association found was between hands/wrists and female gender, this may be due to the small number of females in the final sample (5.7%) compared to the number of males (94.3%).

#### 4.2.2. Age vs. Body Complaints

Besides gender, ageing is another core aspect of the labour market composition in several countries [26]. In recent years, developed countries have experienced an increasing ageing work population [26]. Nonetheless, it is well known that age is one of the factors that cause changes in physical function [56], declining muscle mass and quality and insulin sensitivity [57]. As shown in Figure 3, the number of complaints per individual increased with age for males, except in the 59–68 age group, and females, except for the 39–48 age group. Therefore, evaluating how different body part complaints respond to ageing is of interest.

When considering the effect of ageing, the Cramer’s V test pointed to four moderately significant associations at 0.001 (*p*-value ≤ 0.001), namely shoulders (0.313), hips/thighs (0.244), knees (0.212), and ankles/feet (0.201). Also, three additional weak associations were found at a 0.05 significance level (*p*-value ≤ 0.05), namely neck (0.172), wrist/hands (0.199), and lower back (0.190). These findings are in accordance with the literature providing evidence of an increase in the prevalence of pain with age. Regarding neck pain, Hoy et al. [51] showed that it peaks in the 35–49-year age group and then begins to decline. Similarly, the present study found that workers aged 29–58 were more likely to have neck pain than workers aged 19–28 and 59–68.

It was also found that workers aged 39–68 were more likely to have shoulder pain than workers aged 19–38, in agreement with other studies relating shoulder pain to older age [58]. Additionally, it has long been recognised that the peak prevalence of upper limb disorders is between 35 and 55 years old [59], which resonates with this study’s findings. However, although a significant association was found between lower back pain and age, results were inconclusive, with a higher likelihood of having complaints in the region in the age groups 29–38 and 49–68. Comparatively, Shiri et al. [53] have found that lower back pain slightly decreased with age while lumbar radicular pain increased. Table 7 presents a complete list of associations (significant and non-significant) between gender and body part complaints.

#### 4.2.3. Occupation vs. Body Complaints

The Chi-square test revealed a significant association (*p*-value ≤ 0.05) between job type and the neck, with Cramer’s V effect size (0.145) indicating a weak association between the variables. Additionally, Tuners, Administrative, and Maintenance workers were more likely to have neck pain than Operators. These occupations are exposed to Manual Handling, Awkward Postures, and Repetitive Movements at different levels. Tuners, for instance, must inspect products under a microscope and adjust machines to meet operational requirements. Likewise, workers in the maintenance sector perform various tasks, from cleaning to general repairs, and may be required to maintain a particular neck position for a prolonged time. Also, Administrative workers may report neck pain because of inadequate adjustment of displays and workstations, favouring inadequate postures.

In the literature, several risk factors have been associated with neck pain, such as sedentary behaviour [60], poor self-assessed health, poor psychological conditions, and a prior history of neck or lower back pain [50,61]. Hoy et al. [51] have also found evidence suggesting that “occupation, headaches, emotional problems, low job satisfaction, sedentary work postures, a poor physical work environment (e.g., poor keyboard or mouse position), ethnicity and smoking may be associated with the onset of neck”. Soft tissue disorders of the neck and upper limb have also been strongly associated with “jobs involving prolonged abnormal postures, abnormally high forces or frequent repetition” [59].

Maimaiti et al. [11], when investigating the prevalence of cervical MSD and its relationship with individual, physical, and psychosocial factors among electronics assembly workers, have shown that the prevalence of cervical MSD was higher in workers with longer job tenure. This finding is in accordance with the results of this study, where the average years of service performing the same job function was 16.5 ± 11.2 years.

Concerning lower back problems, a Chi-square test revealed a significant association (*p*-value ≤ 0.001) between job type and developing trouble in the lower back, with Cramer’s V effect size (0.148) indicating a weak association between the variables. Results show that Tuners and Administrative roles were more likely to have lower back problems than Operators and Maintenance roles. Considering Administrative roles, lower back pain has already been reported in occupations that require prolonged sitting [62,63,64]. Likewise, lifting heavy weights is also a high-risk factor for developing back pain [10]. Interestingly, within the established occupation categories, Operators were responsible for lifting heavier weights but reported less back pain than Tuners and Administrative workers. In this case, other risk factors, such as long-time standing [65] and awkward postures [66], may have been more significant. Other factors include gender and age [53], as well as other psychosocial and psychological factors [67].

A significant association (*p*-value ≤ 0.001) between job type and knee trouble was also found. The Cramer’s V effect size (0.187) indicated a weak association between the variables. Accordingly, tuners, administrative, and maintenance roles were found to be more likely to have knee complaints than operators. Knee osteoarthritis, one of the causes of knee pain, has been linked to the direct cumulative damage from repeated occupational exposure to kneeling, bending, or squatting [68]. Other factors linked to knee osteoarthritis/pain include age [68], sedentarism [69], gender [70] and obesity [68]. Therefore, although Operators were also exposed to repetitive movements, they could be less likely to report knee pain because of these different factors.

In addition to knees, neck, and lower back, the analysis of job-related body part complaints revealed a statistically significant association at 0.001 (*p*-value ≤ 0.001) for ankles/feet with a moderate Cramer’s V effect size (0.261). Tuners were more likely to have ankle/foot complaints than workers in Administrative, Maintenance and Operator roles. Risk factors experienced by Tuners include extended time standing, a determinant already reported in the literature [54]. Nevertheless, different issues can cause subjects to experience troubles in ankles/feet, such as footwear choice [12] and foot alignment [71]. Table 8 summarises all associations between occupation and body part complaints.

Even though these were the only significant associations found in this study (Table 8), several other works have provided evidence for the association between job tasks and body part complaints. In relation to wrists/hands, Palmer et al. [72] concluded that “there is a substantial body of evidence that prolonged and highly repetitious flexion or extension of the wrist materially increases the risk of carpal tunnel syndrome, especially when allied with a forceful grip”. Likewise, prolonged use of handheld powered vibratory tools has also been associated with an increased risk of carpal tunnel syndrome [15]. Concerning the use of a computer keyboard and mouse linked to Administrative functions, Thomsen et al. [73] and Mediouni et al. [74] concluded that there was insufficient epidemiological evidence that computer work causes carpal tunnel syndrome. However, the authors determined that some specific work settings involving computer-mouse use may be associated with developing the syndrome. More recently, Feng et al. [75] reported a high prevalence of work-related, clinically confirmed carpal tunnel syndrome symptoms in young Chinese office workers. Their study also linked frequent pain, intensive computer use, and skipping breaks to an increased risk of wrist and hand symptoms.

As for the hip, Harris and Coggon [13] concluded that heavy manual work and occupations such as farmer or construction worker offer elevated risks for hip issues, especially in men. On the other hand, shoulder problems have often been related to occupational sedentary behaviour [60], manual handling [76,77], working above shoulder height [78], repetitive work [79,80], vibration, and awkward postures [55].

Regarding the 10.5% of individuals who have 31 or more years of service (57.4 ± 4.1 years old), the legal retirement age in Portugal is currently 66 years and four months [81]. However, the social protection regime in force allows some flexibility in choosing the retirement age, granting bonuses or reductions in pension amounts to individuals who decide to retire at a lower or higher age, respectively. The minimum age allowed for access to a retirement pension (with reduction) is 55 years old, as long as individuals have completed a contributory career of at least 30 whole years.

#### 4.2.4. BMI vs. Body Complaints

No significant association was found between BMI and body part complaints. Nevertheless, other studies have already provided evidence that confirms the association between BMI and obesity in certain conditions. For instance, Nordstrom et al. [82] have found that every one-unit increase in weight increases the risk of carpal tunnel syndrome by 8%. Also, people with a higher BMI are more likely to be affected by hip osteoarthritis [13]. Among reasons for this association, there are greater joint stress, and metabolism changes that may lead to a predisposition to osteoarthritis [13]. Additionally, obesity has been identified as one of the individual risk factors for shoulder [2,83], knee [14] and lower back pain [53]. Nonetheless, our findings further align with previous research demonstrating a positive association between BMI and age (Table 5), observed in [84], who conducted an anthropometric study in a manufacturing company in Portugal.

### 4.3. Implications of This Study

This study pioneers the peer-reviewed literature for evaluating the prevalence of MSD symptoms within the sewing needle manufacturing industry in Portugal. The findings have several implications for both occupational health and ergonomics research. Firstly, identifying biomechanical risk factors such as manual handling, awkward postures, and repetitive movements underscores the need for targeted ergonomic interventions. Workers in this industry are exposed to these risk factors to varying degrees, contributing to the high prevalence of pain in different body parts.

Within this industry, the lower back was identified as the most commonly affected area, with 54.2% of workers reporting issues in the past 12 months. This was followed by complaints in the neck (42.2%), shoulders (39.0%), ankles/feet (38.2%), and wrists/hands (35.7%). Additionally, there have been significant associations between occupation and body part complaints. For instance, it was found that Tuners, Administrative, and Maintenance roles were more predisposed to having neck problems. Potential reasons include the following. Tuners are required to sustain static postures while standing during machine tuning. Additionally, a forward inclination of the trunk is required for the worker to tune the machines. It is also noted that some machines may have dimensions that are not adjusted to the anthropometric dimensions of the workers, which forces them into inadequate postures. The same may be true for Administrative workers since the mismatch between the worker and the workstation, particularly the height of the tables, displays, and chairs, may cause MSD complaints. Finally, Maintenance workers have a diversity of tasks in their daily routines but forced postures (e.g., squatting) may significantly contribute to the occurrence of MSD complaints in this occupation. Consequently, these different tasks and complaints indicate the need for job-specific ergonomic assessments and interventions.

Females also exhibited a higher frequency of complaints than males across all body parts except for ankles/feet. They were also 1.54 more likely (i.e., risk probability) to have trouble in their wrists/hands than men, with odds (i.e., odds ratio) 2.15 times greater for females than males. Nonetheless, it is worth noting that less than 5.0% of all workers were females. This distribution points to a significant gender gap in the needle manufacturing industry since the final sample is representative of the broader metallic manufacturing industry in Portugal (CAE Rev. 3, Activities 24 and 25).

The age-related increase in MSD complaints, particularly the associations found with shoulders, hips/thighs, knees, and ankles/feet, reinforces the necessity of considering age as a factor in workplace health strategies. As the workforce ages, tailored interventions that address older workers’ physical and health needs will become increasingly important. Interventions could involve job rotation, adaptive equipment, and age-specific health promotion activities to mitigate the impact of ageing on musculoskeletal health.

As MSD is one of the most prevalent occupational health problems in Portuguese companies [32], this work contributes to the existing literature on the prevalence of MSD symptoms in the country by addressing an industry that had not been investigated before. In this context, current findings echo those of Vieira et al. [85] for the Portuguese footwear industry, which asserted that female workers presented a higher prevalence of MSD symptoms than male workers. Likewise, it identified a significant association between reporting symptoms in different body parts and occupations, as found by Moreira-Silva et al. [86] when investigating 202 blue and white-collar workers in Portugal. The authors also found that MSD symptoms were reported more often by older workers.

Among the main limitations, one can cite the structure of the questionnaire, which included sociodemographic characteristics and one question of the NMQ, excluding relevant factors in the assessment of MSD, such as stress levels, physical activity levels, and alcohol and tobacco use. Also, musculoskeletal injuries were not considered in this study. Additionally, as a cross-sectional study, this research is unable to establish time relationships or track workers’ MSD complaints over time [33]. These limitations point to a future research agenda that includes investigating the organisational, psychosocial, and psychological factors influencing the prevalence of MSD complaints in the same industry. Also, continuously applying the NMQ to monitor the development of MSD symptoms over time will allow the authors to assess the effect of workplace ergonomic improvement. In this regard, access to occupational health data can significantly contribute to the assessment of WMSD.

In summary, this study comprehensively evaluates MSD complaints and associated risk factors in the sewing needle manufacturing industry. The findings advocate for ergonomic interventions that are gender-, age-, and job-sensitive. These implications can guide industry stakeholders and policymakers in developing effective occupational health programs and enhancing the overall well-being of workers in this sector in Portugal.

## 5. Conclusions

This study was the first in the literature to investigate the needle manufacturing industry, contributing to the existing body of literature assessing the prevalence of MSD complaints in Portugal. Furthermore, its considerable sample size of 526 individuals allows it to be representative not only at the company level but also at the national level. Within this industry, the most affected body parts were the lower back, neck, shoulders, ankles/feet, and wrists/hands, all reported by more than one-third of the sample. Female workers were a minority in the surveyed company, comprising less than 5% of the final sample. They were also found to be older than men and have a higher complaint rate than men across all body parts except for ankles/feet.

Tuners presented the highest body complaint rate with 3.3 complaints/worker, followed by Maintenance, Administrative, Operators, and Logistics. Significant associations between job function and body part complaints were found for Tuners and ankles/feet complaints; Administrative, Maintenance, and Tuners and having trouble in the Knees and neck; and Administrative and Tuners having trouble in the lower back. To address MSD complaints comprehensively, the company should adopt a holistic approach that goes beyond changing the physical characteristics of workstations. Implementing policies that promote job rotation, incorporate regular rest breaks, supportive leadership, and encourage healthy lifestyle habits can effectively target prevalent musculoskeletal issues.

In conclusion, this study provides a preliminary exploration of MSD risk factors within the needle industry. While significant associations were identified, and the sample achieved representativeness, inherent limitations in the research design restrict the examination of all potential contributors to MSD development or symptom absence.

## Figures and Tables

**Figure 1 ijerph-21-00996-f001:**
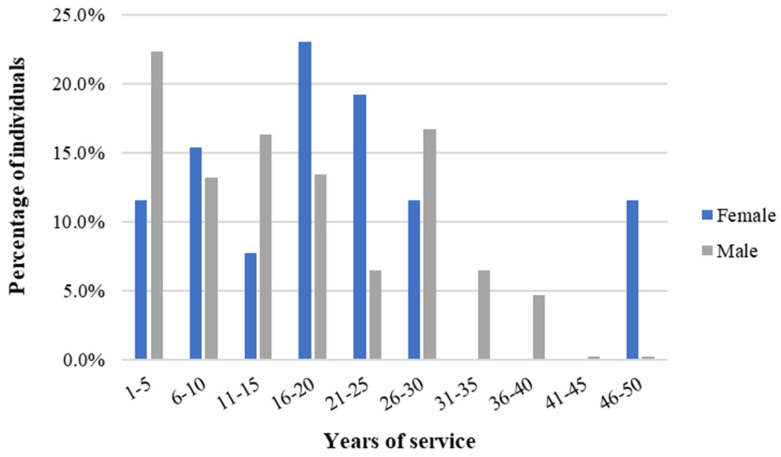
Percentage of individuals per range of years of service in a needle manufacturing industry, Northern Portugal, 2018.

**Figure 2 ijerph-21-00996-f002:**
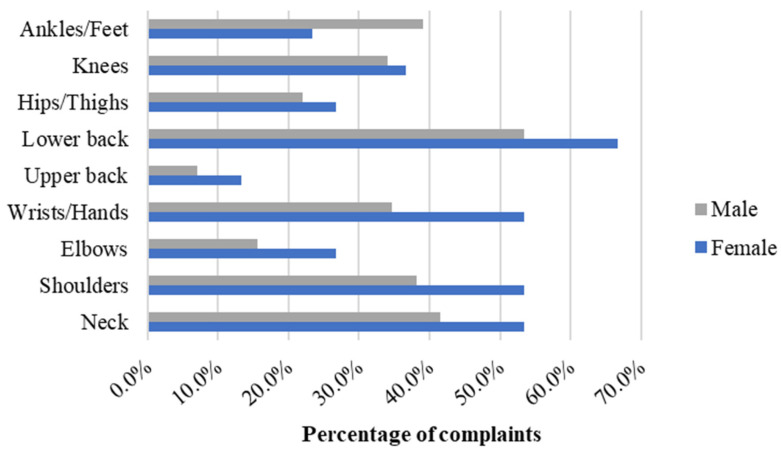
Distribution of workers’ complaints related to body parts by gender in a needle manufacturing industry, Northern Portugal, 2018.

**Figure 3 ijerph-21-00996-f003:**
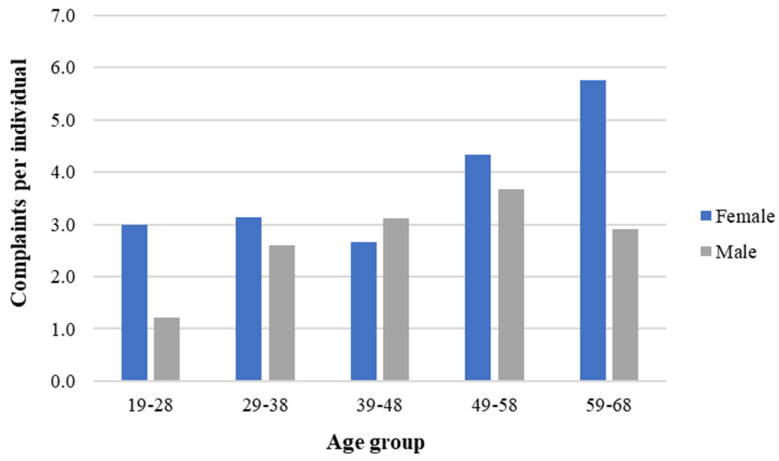
Average number of complaints per worker in different age groups by gender in a needle manufacturing industry, Northern Portugal, 2018.

**Figure 4 ijerph-21-00996-f004:**
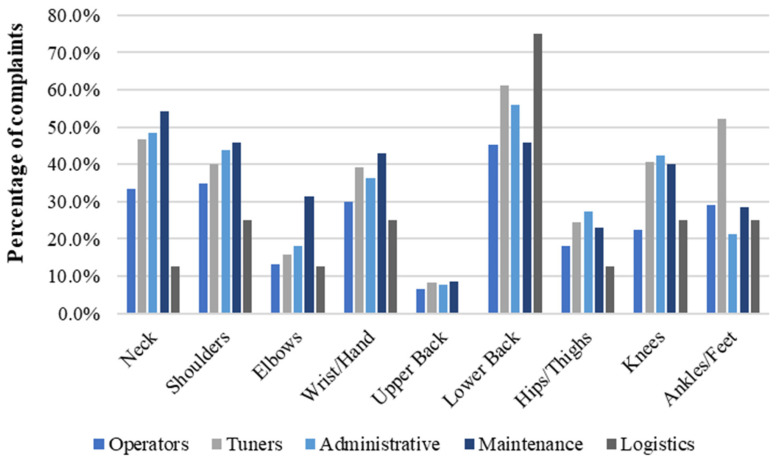
Percentage of complaints per body part and occupation in a needle manufacturing industry, Northern Portugal, 2018.

**Figure 5 ijerph-21-00996-f005:**
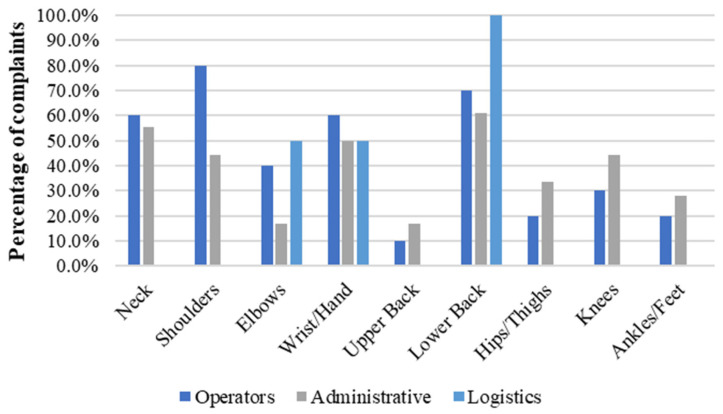
Percentage of complaints per body part among female workers in a needle manufacturing industry, Northern Portugal, 2018.

**Figure 6 ijerph-21-00996-f006:**
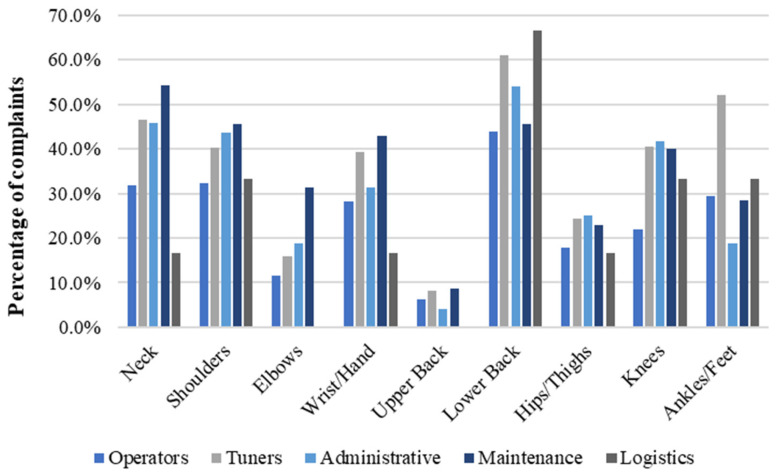
Percentage of complaints per body part among male workers in a needle manufacturing industry, Northern Portugal, 2018.

**Figure 7 ijerph-21-00996-f007:**
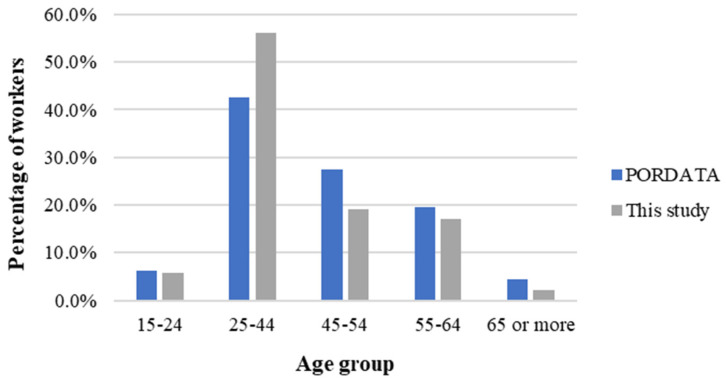
Distribution of workers per age group: This study vs. National profile.

**Table 1 ijerph-21-00996-t001:** Summary of collected data.

Sociodemographic Variables	Body Parts	Have You Had Trouble (Ache, Pain, Discomfort, Numbness) at Any Time During the Last 12 Months? *
Gender Age Job/function Height Weight Body Mass Index (BMI) Shift Years of Service Dominant Handedness	Neck Shoulders Upper Back Elbows Wrists/Hands Low Back Hips/Thighs Knees Ankles/Feet	Yes No

* An affirmative response (ache, pain, discomfort, numbness) within the past twelve months constituted a reported MSD complaint.

**Table 2 ijerph-21-00996-t002:** Frequency and percentage of males and females across different occupations in a needle manufacturing industry, Northern Portugal, 2018.

Occupation	Male	Female	Total
Operators	173 (32.9%)	10 (1.9%)	183 (34.8%)
Tuner	234 (44.5%)	0 (0.0%)	234 (44.5%)
Administrative	48 (9.1%)	18 (3.4%)	66 (12.5%)
Maintenance	35 (6.7%)	0 (0.0%)	35 (6.7%)
Logistics	6 (1.1%)	2 (0.4%)	8 (1.5%)
Total	496 (94.3%)	30 (5.7%)	526 (100%)

**Table 3 ijerph-21-00996-t003:** Homogeneous exposure groups and associated biomechanical risk factors.

Job Title	Task Description	Biomechanical Risk Factors
Operator	Manually handle semi-finished products to supply the production machine (approximately 5–11 kg per batch). Inspection of dimensions under a microscope (self-check for product quality control). Manual handling of loads to storage locations (within a distance of 5 m).	Manual Handling Awkward Postures Repetitive Movements
Tuner	Preparation and adjustment of the machine for operation, involving meticulous fine-tuning that may take 2 to 8 h for each session. This process necessitates the use of tools such as screwdrivers and hex keys. Replacement of machine assemblies weighing between 15 kg and 50 kg. Inspect the product under a microscope to assess the condition of machine calibration and any associated repairs.	Manual Handling Awkward Postures Repetitive Movements
Maintenance	General maintenance activities related to the production line (e.g., metalwork and waste collection) and building maintenance (e.g., floor cleaning, painting, plumbing, and carpentry).	Manual Handling Awkward Postures
Logistics	Performing manual load handling. Loading/unloading trucks with electric pallet trucks and forklifts. Distributing materials across production sectors with forklifts. Handling administrative duties.	Manual Handling Awkward Postures
Administrative	Executing primarily computer-based tasks.	Awkward Postures Repetitive Movements

**Table 4 ijerph-21-00996-t004:** Percentage of individuals per BMI category in a needle manufacturing industry, Northern Portugal, 2018.

Category	Underweight	Normal Weight	Pre-Obesity	Obesity Class I	Obesity Class II	Obesity Class III	Total
BMI Interval	(Below 18.5)	(18.5–24.9)	(25.0–29.9)	(30.0–34.9)	(35.0–39.9)	(Above 40)	
Female	1 (4.2%)	12 (50.0%)	7 (29.2%)	3 (12.5%)	1 (4.2%)	0 (0.0%)	24 (5.4%)
Male	1 (0.2%)	161 (38.3%)	208 (49.5%)	38 (9.0%)	10 (2.4%)	2 (0.5%)	420 (94.6%)
Total	2 (0.5%)	173 (39.0%)	215 (48.4%)	41 (9.2%)	11 (2.5%)	2 (0.5%)	444

**Table 5 ijerph-21-00996-t005:** Cross tabulation between the number of individuals per BMI and age groups in a needle manufacturing industry, Northern Portugal, 2018.

Nutritional Status (BMI)	Age Groups (Years Old)					Total
	19–28	29–38	39–48	49–58	59–68	
Underweight (Below 18.5)	0 (0.0%)	1 (0.8%)	1 (0.8%)	0 (0.0%)	0 (0.0%)	2 (0.5%)
Normal weight (18.5–24.9)	31 (51.7%)	57 (45.2%)	38 (30.2%)	40 (35.7%)	5 (26.3%)	171 (39.1%)
Pre-obesity (25.0–29.9)	25 (41.7%)	55 (43.7)	62 (49.2%)	56 (50.0%)	12 (63.2%)	210 (48.1%)
Obesity class I, II, and III (≥30.0)	4 (6.7%)	13 (10.3%)	19 (15.8%)	16 (14.3%)	2 (10.5%)	54 (12.4%)
Total	60 (100%)	126 (100%)	120 (100%)	112 (100%)	19 (100%)	437

**Table 6 ijerph-21-00996-t006:** Associations between gender and body part complaints among needle manufacturing workers, Northern Portugal, 2018.

Body Parts	Statistics	Interpretation	Odds Ratio
Neck	[χ^2^ (1) = 1.615; *p* > 0.05]	A Chi-square test revealed a non-significant association between gender and developing trouble in the neck.	The odds of having neck trouble are 1.61 times greater for females than males.
Shoulders	[χ^2^ (1) = 2.758; *p* > 0.05]	A Chi-square test revealed a non-significant association between gender and developing shoulder trouble.	The odds of having shoulder trouble are 1.85 times greater for females than males.
Elbow	(*p* = 0.123)	Fisher’s exact test revealed a non-significant association between gender and developing elbow trouble.	The odds of having elbow trouble are 1.98 times greater for females than males.
Wrists/Hands *	[χ^2^ (1) = 4.287; *p* < 0.05]	A Chi-square test revealed a significant association between gender and developing trouble in wrists/hands. Although the result is statistically significant, the variables are only weakly associated. Cramer’s V effect size (0.090) indicates a weak association between the variables.	The odds of having wrist/hand trouble are 2.15 times greater for females than males.
Upper Back	(*p* = 0.267)	Fisher’s exact test revealed no significant association between gender and developing upper back trouble.	The odds of having upper back trouble are 2.02 times greater for females than males.
Lower Back	[χ^2^ (1) = 1.997; *p* > 0.05].	A Chi-square test revealed a non-significant association between gender and developing trouble in the lower back.	The odds of having lower back trouble are 1.74 times greater for females than males.
Hips/Thighs	[χ^2^ (1) = 0.360; *p* > 0.05].	A Chi-square test revealed a non-significant association between gender and developing trouble in the hip/thighs.	The odds of having hip/thigh trouble are 1.29 times greater for females than males.
Knees	[χ^2^ (1) = 0.085; *p* > 0.05].	A Chi-square test revealed a non-significant association between gender and developing knee trouble.	The odds of having knee trouble are 1.12 times greater for females than males.
Ankles/Feet	[χ^2^ (1) = 2.983; *p* > 0.05].	A Chi-square test revealed a non-significant association between gender and developing trouble in ankles/feet.	The odds of having ankles/feet trouble are 2.11 times greater for males than females.

* Indicates a significant association at 0.05 level (*p* < 0.05).

**Table 7 ijerph-21-00996-t007:** Associations between age group and body part complaints among needle manufacturing workers, Northern Portugal, 2018.

Body Parts	Statistics	Interpretation	
Neck **	[χ^2^ (4) = 15.306, *p* < 0.001].	A Chi-square test revealed a significant association between age category and developing neck trouble. Cramer’s V effect size (0.172) indicates a weak association between the variables.	Workers aged 29–58 are more likely to have neck complaints than workers aged 19–28 and 59–68.
Shoulders **	[χ^2^ (4) = 50.588, *p* < 0.001].	A Chi-square test revealed a significant association between age category and developing shoulder trouble. Cramer’s V effect size (0.313) indicates a moderate association between the variables.	Workers aged 39–68 are more likely to have shoulder complaints than workers aged 19–38.
Elbow	Expected count less than 5		
Wrists/Hands **	[χ^2^ (4) = 20.388, *p* < 0.001].	A Chi-square test revealed a significant association between age category and developing trouble in writs/hands. Cramer’s V effect size (0.199) indicates a weak association between the variables.	Workers aged 39–68 are more likely to have wrists/hands complaints than workers aged 19–38.
Upper Back	Expected count less than 5		
Lower Back **	[χ^2^ (4) = 18.566, *p* < 0.001]	A Chi-square test revealed a significant association between age category and developing trouble in the lower back. Cramer’s V effect size (0.190) indicates a weak association between the variables.	Workers aged 29–38 and 49–68 are more likely to have lower back complaints than workers aged 19–28 and 39–48.
Hips/Thighs **	[χ^2^ (4) = 30.655, *p* < 0.001]	A Chi-square test revealed a significant association between age category and developing trouble in the hip/thighs. Cramer’s V effect size (0.244) indicates a moderate association between the variables.	Workers aged 49–68 are more likely to have hip/thigh complaints than workers aged 19–48.
Knees **	[χ^2^ (4) = 23.202, *p* < 0.001]	A Chi-square test revealed a significant association between age category and developing knee trouble. Cramer’s V effect size (0.212) indicates a moderate association between the variables.	Workers aged 39–68 are more likely to have knee complaints than workers aged 19–38.
Ankles/Feet **	[χ^2^ (4) = 20.819, *p* < 0.001]	A Chi-square test revealed a significant association between age category and developing trouble in ankles/feet. Cramer’s V effect size (0.201) indicates a moderate association between the variables.	Workers aged 39–68 are more likely to have ankle/foot complaints than workers aged 19–38.

** indicates a significant association at 0.001 level (*p* < 0.001).

**Table 8 ijerph-21-00996-t008:** Significant and non-significant associations between occupation and body part complaints among needle manufacturing workers, Northern Portugal, 2018.

Body Parts	Statistics	Interpretation	
Neck *	[χ^2^ (3) = 10.828; *p* < 0.05]	A Chi-square test revealed a significant association between job type and developing neck trouble. Cramer’s V effect size (0.145) indicates a weak association between the variables.	Tuners, Administrative, and Maintenance roles are more likely to have neck complaints than operators.
Shoulders	[χ^2^ (3) = 2.710; *p* > 0.05].	A Chi-square test revealed a non-significant association between job type and developing shoulder trouble.	Tuners, Administrative, and Maintenance roles are more likely to have shoulder complaints than operators.
Elbow	[χ^2^ (3) = 7.473; *p* > 0.05].	A Chi-square test revealed a non-significant association between job type and developing trouble in the elbow.	Administrative and Maintenance roles are more likely to have elbow complaints than Operators and Turners.
Wrists/Hands	[χ^2^ (3) = 4.646; *p* > 0.05].	A Chi-square test revealed a non-significant association between job type and developing trouble in wrists/hands.	Tuners, Administrative and Maintenance roles are more likely to have wrist/hand complaints than Operators.
Upper Back	Expected count less than 5		
Lower Back **	[χ^2^ (3) = 11.340; *p* < 0.001]	A Chi-square test revealed a significant association between job type and developing trouble in the lower back. Cramer’s V effect size (0.148) indicates a weak association between the variables.	Tuners and Administrative roles are more likely to have lower back complaints than Operators and Maintenance roles.
Hips/Thighs	[χ^2^ (3) = 3.431; *p* > 0.05]	A Chi-square test revealed a non-significant association between job type and developing trouble in the hip/thighs.	Tuners, Administrative and Maintenance people are more likely to have hips/thigh complaints than Operators.
Knees **	[χ^2^ (3) = 18.031; *p* < 0.001]	A Chi-square test revealed a significant association between job type and developing knee trouble. Cramer’s V effect size (0.187) indicates a weak association between the variables.	Tuners, Administrative and Maintenance roles are more likely to have knee complaints than workers than Operators.
Ankles/Feet **	[χ^2^ (3) = 35.225, *p* < 0.001].	A Chi-square test revealed a significant association between job type and developing trouble in ankles/feet. Cramer’s V effect size (0.261) indicates a moderate association between the variables.	Tuners are more likely to have ankle/foot complaints than workers in Administrative, Maintenance and Operator roles.

* Indicates a significant association at 0.05 level (*p* < 0.05); ** Indicates a significant association at 0.001 level (*p* < 0.001).

## Data Availability

The datasets presented in this article are not readily available due to company restrictions on data sharing.

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
