# Peer review of "Exploring Musculoskeletal Complaints in a Needle Manufacturing Industry: A Cross-Sectional Study"

_ijerph, 2024, doi:10.3390/ijerph21080996_

Round 1

Reviewer 1 Report

Comments and Suggestions for Authors

This is a cross-sectional study on musculoskeletal complaints among workers in a needle manufacturing industry. The authors conducted an extensive literature review on the topic to construct the study's context and justification. There is significant interest in analyzing occupational risk factors for these conditions, which frequently incur direct and indirect costs on individual, institutional, and governmental levels. However, the manuscript presents several points that need critical evaluation regarding its merit.

The principal issue identified is the authors' lack of focus on discussing the relationship between ergonomic aspects of work and the association with musculoskeletal complaints. Specifically, there is a gap in the evaluation and risk mapping necessary to suggest, for instance, that administrative work presents a greater risk for wrist/hand disorders compared to work in the tuner. Additionally, the authors do not address organizational aspects (such as only 30 minutes of break in an eight-hour shift) or psychosocial factors, even though these were not evaluated in the study.

Another point of concern is the disproportion between descriptive and analytical results in the manuscript, with most analyses presented only in the supplementary materials. Various figures that merely illustrate the distribution without contributing to the study’s conclusions should be moved to the supplementary materials, along with information on BMI.

Below are specific points that require attention:

a) The title should refer to complaints instead of disorders, as no clinical evaluation was conducted.

b) The manuscript does not indicate whether an ethics committee reviewed the use of human data and if the study adhered to the Declaration of Helsinki.

c) The materials and methods section should conform to STROBE criteria.

d) The format of the reference presentation in line 44 should be verified.

e) Table 1, which details job functions, should be moved to the results section (item 3.4) along with the analyses.

f) The presentation of tables should follow the standard of indicating the number of participants, research location, and year of data collection.

g) Results should not be included in the discussion section, as seen with Figure 7, Table 4, and Table 5.

h) Lines 266 to 271 pertain to methods and should be relocated accordingly.

i) The discussion section should be reviewed to avoid repetition of results.

j) The manuscript should address whether it is unusual for employees to have been with the company for over 30 years and consider Portugal's retirement legislation.

k) The discussion should focus on the study's results rather than including sections of literature review.

l) The limitations of the study should be positioned at the end of the discussion section.

Addressing these points will enhance the clarity and scientific rigor of the manuscript.

Author Response

Dear reviewer, 

Best regards,

Paulo Anacleto

Reviewer 2 Report

Comments and Suggestions for Authors

Dear authors, thank you for the opportunity to get acquainted with your study.

The uniqueness of this study is due to the specificity of the sample - needle workers in Portugal and its representativeness. The strengths of the study are also the clarity of the research methodology, the clarity of the presentation of the results, the quality of the discussion of the results and the clarity of the conclusions.

The following can be noted as comments on the presentation of the research results in the manuscript:

1. Specifying the purpose of the study, indicating all the research hypotheses, relative to which the results are described.

2. Adding information to the introduction regarding existing studies in the field of studying the occupational health of needle workers.

3. Adding information to the research results regarding the statistical significance of differences between groups by length of service, age, and professional groups, which are presented in the Results section. At the moment, this is not clearly presented in the Research Results, and a clear description appears in the Discussion of Results (this needs to be changed).

4. The Discussion of Results presents information regarding the representativeness of the sample, this needs to be moved to the Research Results section. 5. The authors also present new results in the Discussion of Results that were not reflected in the Results section (Tables 4 and 5); they should be moved to Results, and the Discussion of Results section should contain information only on the interpretation of the results and their correlation with other studies.

6. In the Conclusion, it is necessary to add a conclusion regarding the differences in employees of different professional groups (labor functions).

7. Usually, the limitations of the study are described in detail in the Discussion of Results section, and only briefly summarized in the conclusion.

All comments do not reduce the significance of the study results.

After minor revision, the article can be recommended for publication.

Best wishes, reviewer

Comments on the Quality of English Language

The writing style is clear and correct.

Author Response

(The authors gave the same response as above.)

Reviewer 3 Report

Comments and Suggestions for Authors

Please see attached document for comments.

Author Response

(The authors gave the same response as above.)

Round 2

Reviewer 1 Report

Comments and Suggestions for Authors

The presentation of results, which does not follow the technical principles of communication through self-explanatory tables. The authors believe that the description of the sampling, time, and location is necessary only in the methods section, without repetition in tables and figures, which is not usual in academic circles and in relevant journal publications in the field.

There is yet another point regarding the results, with information that should be in the results section being included in the discussion section. This is also not usual.

Author Response

(The authors gave the same response as above.)

Reviewer 3 Report

Comments and Suggestions for Authors

The paper has been substantially revised to address my concerns about terms, and. presumably, addressing comments from other peer reviewers. I approve of how this revision clearly changed the term "disorders" to "complaints" throughout the article body and the title. Two small suggestions for the author's consideration are: be consistent in using socio-demographic and sociodemographic, and two, include "survey'" among the keywords. Neither of these suggestions are needed for my recommendation to publish the article in its present form.    

Author Response

(The authors gave the same response as above.)
